# Characterisation of MRI Indeterminate Breast Lesions Using Dedicated Breast PET and Prone FDG PET-CT in Patients with Breast Cancer—A Proof-of-Concept Study

**DOI:** 10.3390/jpm10040148

**Published:** 2020-09-25

**Authors:** Anmol Malhotra, Sophia Tincey, Vishnu Naidu, Carla Papagiorcopulo, Debashis Ghosh, Peng H. Tan, Fred Wickham, Thomas Wagner

**Affiliations:** Radiology Department, Royal Free Hospital NHS Trust, London NW3 2QG, UK; sophia.tincey@nhs.net (S.T.); vishnu.naidu@nhs.net (V.N.); carla.papagiorcopulo@nhs.net (C.P.); debashis.ghosh@nhs.net (D.G.); peng.tan@nhs.net (P.H.T.); caspar.wickham@nhs.net (F.W.)

**Keywords:** MRI-indeterminate breast lesion, PET-MAMMI, prone PET-CT

## Abstract

Magnetic resonance imaging (MRI) in patients with breast cancer to assess extent of disease or multifocal disease can demonstrate indeterminate lesions requiring second-look ultrasound and ultrasound or MRI-guided biopsies. Prone positron emission tomography-computed tomography (PET-CT) is a dedicated acquisition performed with a breast-supporting device on a standard PET-CT scanner. The MAMmography with Molecular Imaging (MAMMI, Oncovision, Valencia, Spain) PET system (PET-MAMMI) is a true tomographic ring scanner for the breast. We investigated if PET-MAMMI and prone PET-CT were able to characterise these MRI- indeterminate lesions further. A total of 10 patients with breast cancer and indeterminate lesions on breast MRI were included. Patients underwent prone PET-MAMMI and prone PET-CT after injection of FDG subsequently on the same day. Patients then resumed their normal pathway, with the clinicians blinded to the results of the PET-MAMMI and prone PET-CT. Of the MRI-indeterminate lesions, eight were histopathologically proven to be malignant and two were benign. PET-MAMMI and prone PET-CT only were able to demonstrate increased FDG uptake in 1/8 and 0/8 of the MRI-indeterminate malignant lesions, respectively. Of the MRI-indeterminate benign lesions, both PET-MAMMI and prone PET-CT demonstrated avidity in 1/2 of these lesions. Our findings do not support the use of PET-MAMMI to characterise indeterminate breast MRI lesions requiring a second look ultrasound.

## 1. Introduction

Magnetic resonance imaging (MRI) of the breast is principally used as a supplemental tool in the investigation of breast cancer, along with more conventional methods of mammography and ultrasound. Breast MRI is mainly used for women who have been diagnosed with breast cancer to help accurately measure cancer size and position, look for other tumours in the breast and to check for tumours in the opposite breast [1,2] For certain women at high risk for breast cancer, MRI may be used as a tool in screening detection of breast cancer [3]. MRI is a better tool at detecting invasive carcinoma than full field digital mammography or digital breast tomosynthesis. MRI also has the benefit of having no radiation dose [4].

The use of MRI is not without limitations, however, as it is known to occasionally provide false positive results, which mean more tests and/or biopsies for the patient. As the number of MRI studies being performed is increasing, more indeterminate MRI-detected lesions are being identified that were not initially found by examination or conventional imaging, being classified as BIRADS 3/4a. Since MRI has high sensitivity but lower specificity, these patients require further investigation with second look ultrasound and biopsy, which can be time consuming, expensive and anxiety provoking for the patient [5]. Furthermore, management of these MRI-detected indeterminate lesions can be controversial, and there is no clear guidance as how to manage this subgroup of patients. 

In several published studies indeterminate MRI-detected lesions have been found in 11–29% of patients, with around 3–39% of these being found to be malignant [5,6,7,8]. 

FDG positron emission tomography-computed tomography (PET-CT) has now become standard in the management of many cancer types with a wide range of clinical indications [9]. 

In the UK, ^18^ F-FDG PET CT is becoming recognised as the most accurate imaging modality for the detection of metastatic breast disease recurrence, particularly in the detection and definition of small volume nodal disease as well as bony metastases. Currently, ^18^ F-FDG PET CT is used mainly for patients with equivocal imaging or clinical findings, providing a definitive assessment with regards to the presence or absence of active recurrent metastatic breast cancer [10]. Furthermore, it is useful in further assessing multifocal disease, recurrent disease in patients with dense breasts, and differentiating treatment-induced brachial plexopathy from tumour infiltration in those patients with normal or equivocal MRI findings [10]. 

PET-CT is not optimal for evaluation of the breasts due to supine positioning and low resolution (7–8 mm) which limits detection of sub-centimetre cancers [11]. One way of improving the quality of images is to scan patients prone with the help of a supporting device between the breasts (prone PET-CT) [12].

Another technology currently being investigated in breast cancer is a dedicated breast PET, which uses PET technology in devices specifically suited to look for abnormalities within the breasts. It has high sensitivity for invasive breast cancer, and has been validated in women with newly diagnosed breast cancer to depict disease extent as an alternative to breast MRI [13]. The MAMmography with Molecular Imaging (MAMMI, Oncovision, Valencia, Spain) PET system is a true tomographic ring scanner where patients are scanned in the prone position. The breast hangs pendulous without compression within a ring of detectors that move up and down the breast [11].

## 2. Materials and Methods

All subjects gave their informed consent for inclusion before they participated in the study. The study was conducted in accordance with the Declaration of Helsinki, and the protocol was approved by the London-Bloomsbury Ethics Commitee on 15/06/2016 (REC reference: 191866, IRAS Project ID:** 191866).

All breast MRI were performed using a standard full breast MRI protocol with contrast administration in patients with known new breast cancer diagnosis for local staging. Single reader was performed by radiologists with between 8–18 years of breast MRI experience. 

Inclusion and exclusion criteria are described in Table 1.

### 2.1. PET-CT

Patients underwent 18F-FDG-PET/CT using a Biograph mCT PET/CT system (Siemens, Munich, Germany) following intravenous administration of 18F-FDG and a 60-min uptake period. The activity of FDG was calculated using a weight-based formula according to departmental policy, up to a maximum of 400 MBq. Before 18F-FDG administration, patients fasted for at least 6 h, with blood glucose verified to be 11 mmol/L or less. Unenhanced CT imaging was performed in line with standard protocols from skull base or skull vertex to mid-thigh using automatic exposure control (Siemens CARE Dose4D and CARE kV) with a reference of 65 mAs and 120 kV. PET imaging was performed over the same area at 3 min per bed position, depending on activity injected and body mass index (BMI), with a bed position overlap of 42%. Reconstruction methods included time of flight and point spread function, with two iterations, 21 subsets, and a Gaussian 2 mm full-width half maximum filter.

Supine PET-CT was a standard acquisition with the patient lying flat on their back and being scanned from skull base to thighs.

For the prone PET-CT acquisition the patients lay on their front with a triangular-shaped supporting device allowing the breasts to hang. Prone scanning was performed immediately following the standard supine acquisition and scan time was 12 min.

Following the PET-CT scan patients underwent the PET-MAMMI acquisition.

### 2.2. PET-MAMMI

Both breasts were scanned at 4 min per detector position. The number of detector positions required per breast varied from 2 to 4, depending on the breast length. Images were reconstructed in 3D with 1 mm^3^ voxels using a maximum likelihood expectation maximisation algorithm, including random, scatter and attenuation correction using 12 iterations.

Patients then resumed their normal pathway with the clinicians blinded to the results of the PET-CT and PET-MAMMI.

Results of PET-CT prone and supine and of PET-MAMMI were compared with breast MRI, second-look ultrasound, biopsy and surgical pathology results.

## 3. Results

A total of 11 patients were recruited into the PET-MAMMI study. One patient was excluded as this patient had an index left breast lesion with an indeterminate linear 7 mm focus in the left breast with type 2 enhancement. The advice was for MDM discussion. The patient then had a left mastectomy as per patient choice. All 10 patients had an index lesion and MRI-indeterminate lesion(s). The mean age of the patients was 55.5 (median of 57 with a range of 44–70). 

To obtain tissue specimens, 5/10 patients underwent ultrasound-guided biopsy, 3/10 patients underwent MRI guided biopsy, 1/10 had a wire-guided wide local excision (WLE) and 1/10 patients had a skin-sparing mastectomy (see Figure 1). 

Out of the 10 MRI indeterminate lesions, eight were biopsy-proven malignant (ductal carcinoma in situ (n = 4), invasive ductal carcinoma (n = 1) and invasive lobular carcinoma (n = 3)). The remaining lesions were benign fibroadenomas (n = 2) (see Figure 2 and Table 2). 

PET-MAMMI and prone PET-CT demonstrated increased FDG uptake in 1/8 and 0/8 of the MRI-indeterminate malignant lesions, respectively. The patient with positive PET-MAMMI findings had invasive ductal carcinoma (see Figure 3 and Figure 4). 

PET-MAMMI and prone PET-CT both detected 1/2 of the MRI indeterminate lesions proven to be fibroadenomas (see Figure 3 and Figure 5). 

The mean SUVmax of PET-MAMMI and prone PET-CT were 2 and 1.4, respectively. For the malignant lesions, the mean PET-MAMMI SUVmax was 1.1 (range 0–9) and mean prone PET-CT SUVmax was 1.2 (range 0–3.7). For the benign lesions, the mean PET-MAMMI SUVmax was 5.5 (range 0–11) and mean prone PET-CT SUVmax was 2 (range 0–4). The sensitivity of PET-MAMMI in correctly characterising indeterminate lesions as malignant disease was 12.5% (Table 3). The negative predictive value for PET-MAMMI was 12.5% (95% CI 3.4%–36.9%). Prone PET-CT failed to detect any of the indeterminate lesions proven to be malignant. The negative predictive value of prone PET-CT was 11.1% (95% CI 3.03%–33.3%). 

## 4. Discussion

Dedicated breast PET has been shown by others to demonstrate high sensitivity for breast lesions and has shown promise for evaluating extent of disease in patients diagnosed with breast cancer [13]. This study is the first to evaluate the use of PET-MAMMI and prone PET-CT in the evaluation of breast MRI indeterminate lesions.

In our cohort of patients, we found that PET-MAMMI was not able to reliably characterise MRI-indeterminate lesions. The majority of malignant breast lesions that presented as indeterminate on breast MRI did not demonstrate FDG-avidity on PET-MAMMI and PET-CT. Conversely, benign fibroadenomata that presented as indeterminate lesions on breast MRI were FDG-avid. 

Furthermore, lesions lying close to the pectoralis muscle were not reliably imaged with this technique giving rise to another significant diagnostic limitation. This has been described previously by Teixeira et al. [14]. 

Our study is limited by its small sample size.

Index lesions were well imaged on PET-CT and PET-MAMMI but there is no evidence that findings from breast images of PET-CT or from PET-MAMMI change management.

Other studies investigating the value of dedicated breast PET did find clinical usefulness for the evaluation of the extent of disease [14,15,16].

FDG PET-CT has been used to assess response to treatment in patients undergoing neoadjuvant chemotherapy [17] and PET MAMMI could have a role in this clinical indication.

## 5. Conclusions

Our findings do not support the use of PET MAMMI to characterise indeterminate breast MRI lesions requiring imaging and biopsy. Breast MRI is exquisitely sensitive in the detection of invasive breast cancer [18]. The specificity in this study is high and may reflect local breast MRI expertise.

## Figures and Tables

**Figure 1 jpm-10-00148-f001:**
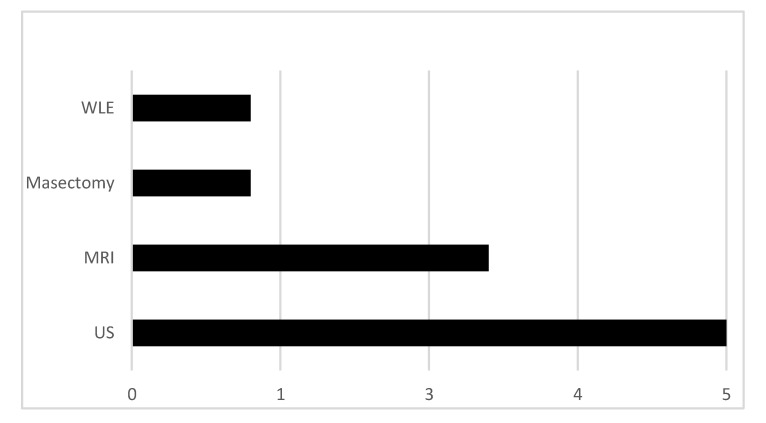
Methods used to sample tissue from magnetic resonance imaging (MRI)-indeterminate lesions.

**Figure 2 jpm-10-00148-f002:**
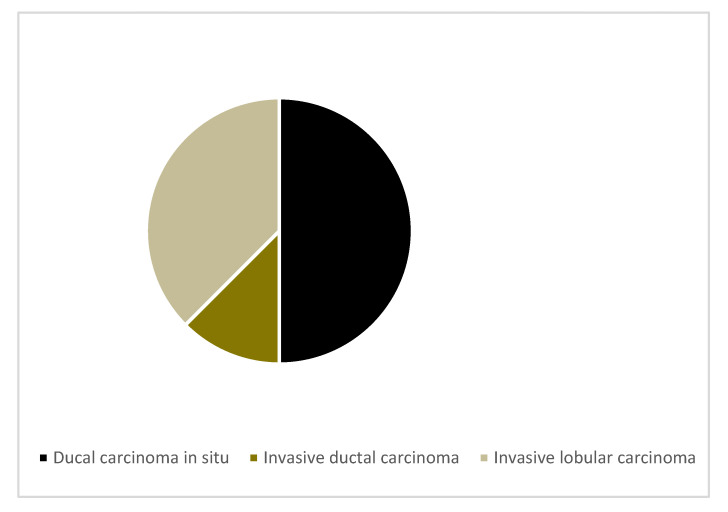
Malignant histological subtypes.

**Figure 3 jpm-10-00148-f003:**
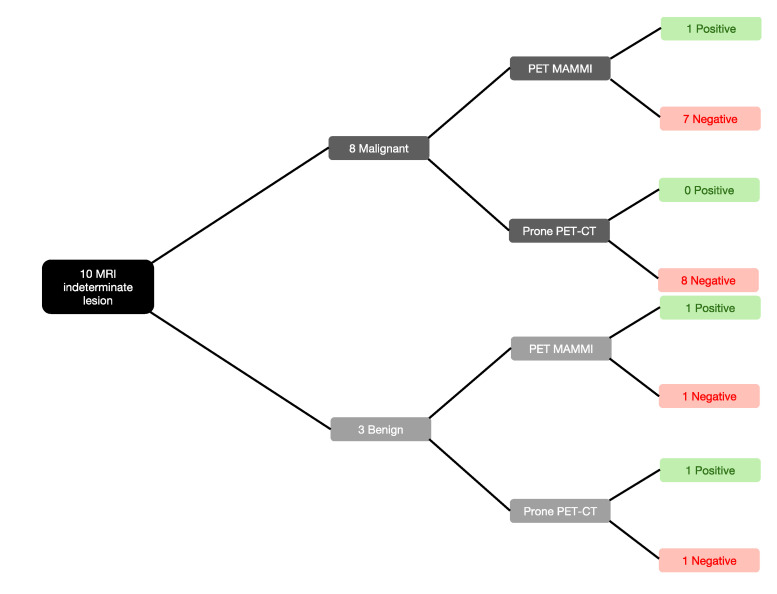
PET-MAMmography with Molecular Imaging (MAMMI) and prone positron emission tomography-computed tomography (PET-CT) performance against malignant and benign biopsy-proven lesions.

**Figure 4 jpm-10-00148-f004:**
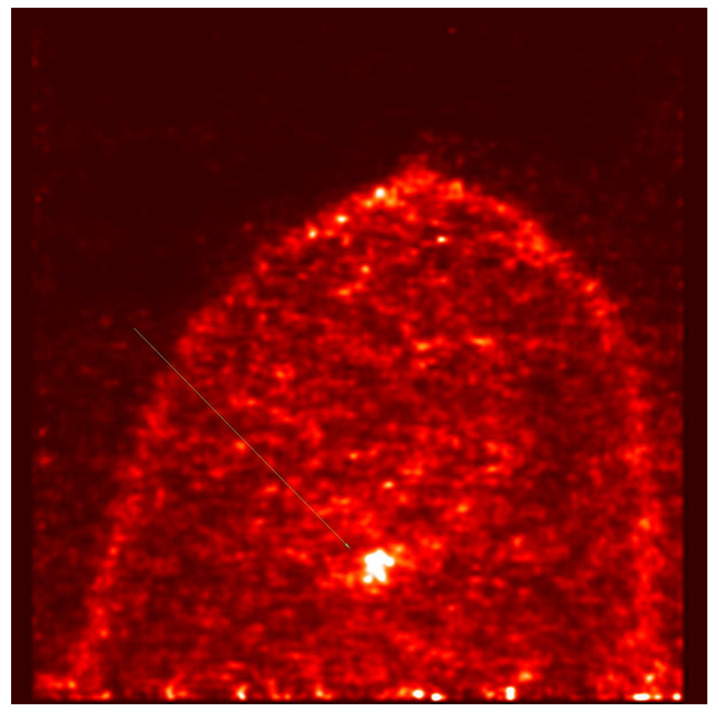
PET- MAMMI, focus of intense uptake in the left breast in a patient with biopsy proven invasive ductal carcinoma.

**Figure 5 jpm-10-00148-f005:**
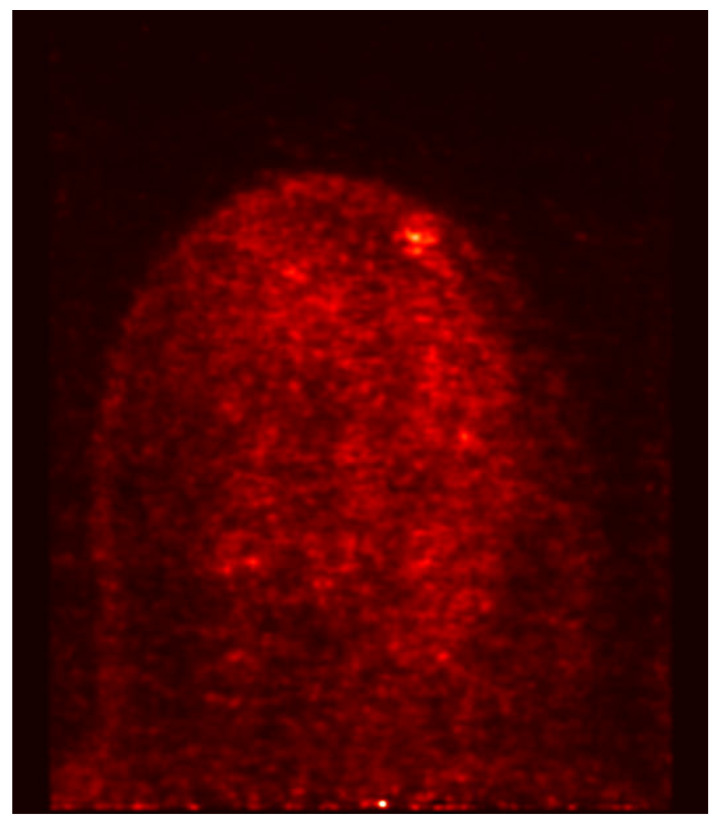
PET- MAMMI, focus of moderate uptake in the left breast in a patient with biopsy proven fibroadenoma.

**Table 1 jpm-10-00148-t001:** Inclusion and exclusion criteria.

Inclusion Criteria	Exclusion Criteria
Female	Pregnant females, planning pregnancy or breastfeeding
Age over 18 (no upper limit)	Concurrent and/or recent involvement in other research that is likely to interfere with the intervention within 3 months prior to study enrolment
Not pregnant	Inability to lie flat or undergo the tests
Not breastfeeding	Any co-morbidities or conditions which in the opinion of the clinical team means that the patient should be excluded
Indeterminate breast lesion on MRI requiring a second-look ultrasound	
Ability to lie still for up to 30 min prone and supine	
Females of childbearing potential have a negative pregnancy test within 7 days prior to being registered. Participants are considered not of childbearing potential if they are surgically sterile (i.e., they have undergone a hysterectomy, bilateral tubal ligation, or bilateral oophorectomy) or they are postmenopausal	
Willing and able to provide written informed consent	

**Table 2 jpm-10-00148-t002:** Size ranges of the MRI indeterminate lesions.

	Number	Size Range (mm)	Lesion (mm)	Location of Indeterminate Lesion with Respect to Index Lesion (Contralateral vs. Ipsilateral)
Fibroadenoma	2	7–12	7	Ipsilateral
			12	Ipsilateral
ILC	3	8–52	8	Ipsilateral
			9	Ipsilateral- related to known disease
			52	Ipsilateral- related to known disease
DCIS	4	5–20	5	Ipsilateral
			6	Ipsilateral- related to known disease
			6	Ipsilateral
			20	Contralateral
IDC	1	11	11	Ipsilateral

**Table 3 jpm-10-00148-t003:** Diagnostic accuracy measures (with 95% confidence intervals) for PET-MAMMI and prone PET-CT (statistics derived from https://www.medcalc.org/calc/diagnostic_test.php). PET-MAMMI: TP = 1, FN = 7, FP = 1, TN = 1, meaning a = 1, b = 7, c = 1, d = 1 Prone PET-CT: TP = 0, FN = 8, FP = 1, TN = 1, meaning a = 0, b = 8, c = 1, d = 1.

	PET-MAMMI	Prone PET-CT
Sensitivity	12.5% (95% CI: 0.3–52.7%)	0% (95% CI: 0–36.9%)
Specificity	50% (95% CI: 1.3–99%)	50% (95% CI: 1.3–98.7%)
Positive predictive value (PPV)	50% (95% CI: 9.1–90.9%)	0
Negative predictive value (NPV)	12.5% (95% CI: 3.4–36.9%)	11.1% (95% CI: 3–33.3%)
Accuracy	20% (95% CI: 2.6–55.6%)	10% (95% CI: 0.3–44.5%)

Of note, 1/10 of the biopsy-proven malignant lesions was too close to the chest wall to be reliably identified by PET-MAMMI.

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
