# Peer review of "Characterisation of MRI Indeterminate Breast Lesions Using Dedicated Breast PET and Prone FDG PET-CT in Patients with Breast Cancer—A Proof-of-Concept Study"

_jpm, 2020, doi:10.3390/jpm10040148_

Round 1
Reviewer 1 Report
The paper ”Characterisation of MRI indeterminate breast lesions using dedicated breast PET and prone FDG PET-CT in patients with breast cancer” by Malhotra et al. illuminates an interesting idea to further characterize MRI-indeterminate lesions in breast cancer by means of PET MAMMI and prone PET-CT. The project is well described, the results are nicely and pedagogically displayed, and the conclusions are balanced; however, both Methods and Results can be further improved as indicated below.
Major:
The study is so small in sample size that it should be indicated as such in its title. I propose to add “ – a proof-of-concept study” at the end of it
L.96-97: A Statistics section is missing, please add. It must describe how descriptive statistics in terms of figures and numbers with respect to continuous [mean and range] and categorical variables [frequencies and percentages] were used and how 95% confidence intervals for binomial proportions were derived (exact Clopper-Pearson type confidence intervals for binomial proportions). Moreover, the analytical package used for the generation of figures and tables must be specified at the end of this section. Inference statistics were not and should not be applied, taking the findings and the sample size into consideration.
L.112, Figure 2: “n=4”, “n=3”, and “n=1” should be added – either in the legend (e.g. “Ducal carcinoma in situ (n=4)”) or appropriately placed next to the respective pies in the pie chart.
L.130: Replace “disease was 12.5%.” by “disease was 12.5% (Table 2).” and add a new Table 2 that shows the usual accuracy parameters (sensitivity, specificity, positive and negative predictive value, accuracy) including respective 95% confidence intervals for both PET MAMMI and prone PET-CT. Though the sample size is small, point estimates enriched by respective 95% confidence intervals do always clearly indicate the uncertainty behind the point estimates and improve the presentation of results. I attached a proposal of how this table could look like for your convenience. Depending on whether you actually used a statistical analysis package like SAS, Stata, or SPSS, you may alternatively make use of an online calculator for the point estimates and exact binomial confidence intervals of the accuracy parameters like this one: https://www.medcalc.org/calc/diagnostic_test.php (see attached table proposal for further information).
L.25-26 and l.159: delete the notion of “further research is required”. Apparently, the results were that futile that it is difficult to see how the patients benefit from further research into that direction. Alternatively, specify as to how further research could be promising in line 159.
Minor corrections:
Lines 2-4: the title should not be underlined
L.27: delete semicolon at the end of the line
L.37: Replace “MRI is not without its limitations” by “The use of MRI is not without limitations”
L.46: Replace “In several published studies the range of indeterminate MRI-detected lesions has been” by “In several published studies indeterminate MRI-detected lesions have been”
L.101: Delete one of two appearances of ‘patients’ in “All 10 patients included patients”
L.118 and l.120: Figure 3, not 4
L.121, Figure 3: box on the left hand side includes “11 MRI…”, whereas it should read “10 MRI…” due to the exclusion of one patient.
L.135: delete “3.1 Figures”
L.167: add reason of why you would like to acknowledge Ms Amy Pritchard
L.170-216: the references do NOT comply with the format of references in the Journal of Personalized Medicine

Author Response
L2: added ‘a proof of concept study’
L:2-4: title not underlined
L.27: delete semicolon at the end of the line
L26 and L:181: deleted ‘however further research is required’
L.37: Replaced “MRI is not without its limitations” by “The use of MRI is not without limitations”
L.49: Replaced “In several published studies the range of indeterminate MRI-detected lesions has been” by “In several published studies indeterminate MRI-detected lesions have been”
L.112: n=x added alongside indeterminate lesion numbers
L.128: and L.130: Changed to ‘Figure 3’
L128: Statistics table added
L.131: Figure 3: modified to say 10 MRI indeterminate lesions
L.135: deleted “3.1 Figures”
References modified to fit in with guidelines
I hope the statistics table is acceptable- I did indeed use the medicalc as you had suggested. Many thanks!
Please see the modified article with all the required modifications.

Reviewer 2 Report
The authors present an interesting research work on the additional use of hybrid imaging techniques in MRI indeterminate lesions. The overall topic is important. However, the number of patients seems relatively low. The authors do report a limited utility of FDG-PET in indeterminate breast lesions.
Point by point comments:
Abstract
“however, further research is required” -> Please specify which kind of research to answer which question? More patients? Other settings? …?
Introduction
Please add a sentence on the potential benefits of MRI in terms of dose reduction and high comparative effectiveness in patients with dense breasts (eg. Dense Study, NEJM).
P 1 line 35-36
“For certain women at high risk for breast cancer, MRI may 36 be used as a tool in screening detection of breast cancer.“ -> Please provide reference
Materials and Methods
The diagnostic PET modalities are described adequately
How was the status of an indeterminate lesion determined in MRI? Multi-reader? Clinical routine? Please describe this and the MRI protocol adequately (short protocol? …)
However, I would strongly advice to add a section on the patient recruitment criteria, IRB approval and, if applicable, a study diagram (eg. Consort)
Results
please remove headings from all figures and put it in the figure legends
It would support your work a lot if you gave two example patient images (one later proven to show malignancy, and one with a benign indeterminate lesion)
Figure 3: The text in the lighter grey boxes (Benign, ..) cannot be read well. Increase text size and contrast please
P 7 line 365: what das 3.1 figures mean?
Did the patient also receive x-ray based mammography before MRI? What was the result?
Discussion
Relatively short
In your study, the majority of MRI deemed indeterminate lesions were shown to be indeed malignant. This may underline the utility of MRI in this setting. The conclusion may, therefore, also include comments taking this into account and discussing the utility of MRI without PET (?).
Author Response
Abstract
Further research is required has been removed
Introduction
Added few more sentences about MRI benefits.
L35-37: MRI is a better tool at detecting invasive carcinoma than full field digital mammography or digital breast tomosynthesis. MRI also has the benefit of having no radiation dose (4).
P 1 line 35-36
Reference added
Methods
L82: Indeterminate lesion status method added: All breast MRI were performed using a standard full breast MRI protocol with contrast administration in patients with known new breast cancer diagnosis for local staging. Single reader was performed by radiologists with between 8-18 years of breast MRI experience.
L131: Recruitment criteria added
Results
Removed headings from all figures and put it in the figure legends
Colour of figure 3 improved
L201/216: Two patient images added
Discussion
L258: Conclusion added to:
Breast MRI is exquisitely sensitive in the detection of invasive breast cancer (18). The specificity in this study is high and may reflect local Breast MRI expertise.
Please see the attached document/ updated article

Round 2
Reviewer 2 Report
The points brought up have been addressed accordingly.